# RÉNYI FAIR INFERENCE

**Sina Baharlouei**
Industrial and Systems Engineering, USC
`baharlou@usc.edu`

**Maher Nouiehed**
Industrial Engineering and Management, AUB
`mn102@aub.edu.lb`

**Ahmad Beirami**
EECS, MIT
`beirami@mit.edu`

**Meisam Razaviyayn**
Industrial and Systems Engineering, USC
`razaviya@usc.edu`

## ABSTRACT

Machine learning algorithms have been increasingly deployed in critical automated decision-making systems that directly affect human lives. When these algorithms are solely trained to minimize the training/test error, they could suffer from systematic discrimination against individuals based on their sensitive attributes, such as gender or race. Recently, there has been a surge in machine learning society to develop algorithms for *fair* machine learning. In particular, several adversarial learning procedures have been proposed to impose fairness. Unfortunately, these algorithms either can only impose fairness up to linear dependence between the variables, or they lack computational convergence guarantees. In this paper, we use Rényi correlation as a measure of fairness of machine learning models and develop a general training framework to impose fairness. In particular, we propose a min-max formulation which balances the accuracy and fairness when solved to optimality. For the case of discrete sensitive attributes, we suggest an iterative algorithm with theoretical convergence guarantee for solving the proposed min-max problem. Our algorithm and analysis are then specialized to fair classification and fair clustering problems. To demonstrate the performance of the proposed Rényi fair inference framework in practice, we compare it with well-known existing methods on several benchmark datasets. Experiments indicate that the proposed method has favorable empirical performance against state-of-the-art approaches.

## 1 INTRODUCTION

As we experience the widespread adoption of machine learning models in automated decision-making, we have witnessed increased reports of instances in which the employed model results in discrimination against certain groups of individuals – see Datta et al. (2015); Sweeney (2013); Bolukbasi et al. (2016); Angwin et al. (2016). In this context, discrimination is defined as the unwanted distinction against individuals based on their membership to a specific group. For instance, Angwin et al. (2016) present an example of a computer-based risk assessment model for recidivism, which is biased against certain ethnicities. In another example, Datta et al. (2015) demonstrate gender discrimination in online advertisements for web pages associated with employment. These observations motivated researchers to pay special attention to fairness in machine learning in recent years; see Calmon et al. (2017); Feldman et al. (2015); Hardt et al. (2016); Zhang et al. (2018); Xu et al. (2018); Dwork et al. (2018); Fish et al. (2016); Woodworth et al. (2017); Zafar et al. (2017; 2015); Pérez-Suay et al. (2017); Bechavod & Ligett (2017); Liao et al. (2019).

In addition to its ethical standpoint, equal treatment of different groups is legally required by many countries Act. (1964). Anti-discrimination laws imposed by many countries evaluate fairness by notions such as disparate treatment and disparate impact. We say a decision-making process suffers from disparate treatment if its decisions discriminate against individuals of a certain protected group based on their sensitive/protected attribute information. On the other hand, we say it suffers from disparate impact if the decisions adversely affect a protected group of individuals with certain sensitive attribute – see Zafar et al. (2015). In simpler words, disparate treatment is intentional

discrimination against a protected group, while the disparate impact is an unintentional disproportionate outcome that hurts a protected group. To quantify fairness, several notions of fairness have been proposed in the recent decade Calders et al. (2009); Hardt et al. (2016). Examples of these notions include *demographic parity*, *equalized odds*, and *equalized opportunity*.

Demographic parity condition requires that the model output (e.g., assigned label) be independent of sensitive attributes. This definition might not be desirable when the base ground-truth outcome of the two groups are completely different. This shortcoming motivated the use of *equalized odds* notion Hardt et al. (2016) which requires that the model output is conditionally independent of sensitive attributes given the ground-truth label. Finally, *equalized opportunity* requires having equal false positive or false negative rates across protected groups.

Machine learning approaches for imposing fairness can be broadly classified into three main categories: pre-processing methods, post-processing methods, and in-processing methods. Pre-processing methods modify the training data to remove discriminatory information before passing data to the decision-making process Calders et al. (2009); Feldman et al. (2015); Kamiran & Calders (2010; 2009; 2012); Dwork et al. (2012); Calmon et al. (2017); Ruggieri (2014). These methods map the training data to a transformed space in which the dependencies between the class label and the sensitive attributes are removed Edwards & Storkey (2015); Hardt et al. (2016); Xu et al. (2018); Sattigeri et al. (2018); Raff & Sylvester (2018); Madras et al. (2018); Zemel et al. (2013); Louizos et al. (2015). On the other hand, post-processing methods adjust the output of a trained classifier to remove discrimination while maintaining high classification accuracy Fish et al. (2016); Dwork et al. (2018); Woodworth et al. (2017). The third category is the in-process approach that enforces fairness by either introducing constraints or adding a regularization term to the training procedure Zafar et al. (2017; 2015); Pérez-Suay et al. (2017); Bechavod & Ligett (2017); Berk et al. (2017); Agarwal et al. (2018); Celis et al. (2019); Donini et al. (2018); Rezaei et al. (2019); Kamishima et al. (2011); Zhang et al. (2018); Bechavod & Ligett (2017); Kearns et al. (2017); Menon & Williamson (2018); Alabi et al. (2018). The Rényi fair inference framework proposed in this paper also belongs to this in-process category.

Among in-processing methods, many add a regularization term or constraints to promote statistical independence between the classifier output and the sensitive attributes. To do that, various independence proxies such as mutual information Kamishima et al. (2011), false positive/negative rates Bechavod & Ligett (2017), equalized odds Donini et al. (2018), Pearson correlation coefficient Zafar et al. (2015; 2017), Hilbert Schmidt independence criterion (HSIC) Pérez-Suay et al. (2017) were used. As will be discussed in Section 2, many of these methods cannot capture nonlinear dependence between random variables and/or lead to computationally expensive algorithms. Motivated by these limitations, we propose to use Rényi correlation to impose several known group fairness measures. Rényi correlation captures nonlinear dependence between random variables. Moreover, Rényi correlation is a normalized measure and can be computed efficiently in certain instances.

Using Rényi correlation coefficient as a regularization term, we propose a min-max optimization framework for fair statistical inference. In particular, we specialize our framework to both classification and clustering tasks. We show that when the sensitive attribute(s) is discrete (e.g., gender and/or race), the learning task can be efficiently solved to optimality, using a simple gradient ascent-descent approach. We summarize our contributions next:

- We introduce Rényi correlation as a tool to impose several notions of group fairness. Unlike Pearson correlation and HSIC, which only capture linear dependence, Rényi correlation captures any statistical dependence between random variables as zero Rényi correlation implies independence. Moreover, it is more computationally efficient than the mutual information regularizers approximated by neural networks.

- Using Rényi correlation as a regularization term in training, we propose a min-max formulation for fair statistical inference. Unlike methods that use an adversarial neural network to impose fairness, we show that in particular instances such as binary classification, or discrete sensitive variable(s), it suffices to use a simple quadratic function as the adversarial objective. This observation helped us to develop a simple multi-step gradient ascent descent algorithm for fair inference and guarantee its theoretical convergence to first-order stationarity.

- Our Rényi correlation framework leads to a natural fair classification method and a novel fair $K$-means clustering algorithm. For $K$-means clustering problem, we show that sufficiently

large regularization coefficient yields perfect fairness under disparate impact doctrine. Unlike the two-phase methods proposed in Chierichetti et al. (2017); Backurs et al. (2019); Rösner & Schmidt (2018); Bercea et al. (2018); Schmidt et al. (2018), our method does not require any pre-processing step, is scalable, and allows for regulating the trade-off between the clustering quality and fairness.

## 2  RÉNYI CORRELATION

The most widely used notions for group fairness in machine learning are demographic parity, equalized odds, and equalized opportunities. These notions require (conditional) independence between a certain model output and a sensitive attribute. This independence is typically imposed by adding fairness constraints or regularization terms to the training objective function. For instance, Kamishima et al. (2011) added a regularization term based on mutual information. Since estimating mutual information between the model output and sensitive variables during training is not computationally tractable, Kamishima et al. (2011) approximates the probability density functions using a logistic regression model. To have a tighter estimation, Song et al. (2019) used an adversarial approach that estimates the joint probability density function using a parameterized neural network. Although these works start from a well-justified objective function, they end up solving approximations of the objective function due to computational barriers. Thus, no fairness guarantee is provided even when the resulting optimization problems are solved to global optimality in the large sample size limit.

A more tractable measure of dependence between two random variables is the Pearson correlation. The Pearson correlation coefficient between the two random variables $A$ and $B$ is defined as $\rho_P(A, B) = \frac{\text{Cov}(A,B)}{\sqrt{\text{Var}(A)}\sqrt{\text{Var}(B)}}$, where $\text{Cov}(\cdot, \cdot)$ denotes the covariance and $\text{Var}(\cdot)$ denotes the variance. The Pearson correlation coefficient is used in Zafar et al. (2015) to decorrelate the binary sensitive attribute and the decision boundary of the classifier. A major drawback of Pearson correlation is that it only captures linear dependencies between random variables. In fact, two random variables $A$ and $B$ may have strong dependence but have zero Pearson correlation. This property raises concerns about the use of the Pearson correlation for imposing fairness. Similar to the Pearson correlation, the HSIC measure proposed in Pérez-Suay et al. (2017) may be zero even if the two variables have strong dependencies. While universal Kernels can be used to alleviate this issue, they could arrive at the expense of computational intractability. In addition, HSIC is not a normalized dependence measure Gretton et al. (2005b;a) which raises concerns about the appropriateness of using it as a measure of dependence.

In this paper, we suggest to use Hirschfeld-Gebelein-Rényi correlation Rényi (1959); Hirschfeld (1935); Gebelein (1941) as a dependence measure between random variables to impose fairness. Rényi correlation, which is also known as maximal correlation, between two random variables $A$ and $B$ is defined as

$$\rho_R(A, B) = \sup_{f,g} \mathbb{E}[f(A)g(B)] \qquad \text{s.t.} \quad \mathbb{E}[f(A)] = \mathbb{E}[g(B)] = 0, \quad \mathbb{E}[f^2(A)] = \mathbb{E}[g^2(B)] = 1,$$
(1)

where the supremum is over the set of measurable functions $f(\cdot)$ and $g(\cdot)$ satisfying the constraints. Unlike HSIC and Pearson correlation, Rényi correlation captures higher-order dependencies between random variables. Rényi correlation between two random variables is zero if and only if the random variables are independent, and it is one if there is a strict dependence between the variables Rényi (1959). These favorable statistical properties of $\rho_R$ do not come at the price of computational intractability as opposed to other measures such as mutual information. In fact, as we will discuss in Section 3, $\rho_R$ can be used in a computationally tractable framework to impose several group fairness notions.

## 3  A GENERAL MIN-MAX FRAMEWORK FOR RÉNYI FAIR INFERENCE

Consider a learning task over a given random variable $\mathbf{Z}$. Our goal is to minimize the average inference loss $\mathcal{L}(\cdot)$ where our loss function is parameterized with parameter $\boldsymbol{\theta}$. To find the optimal

value of parameter $\boldsymbol{\theta}$ with the smallest average loss, we solve the following optimization problem

$$\min_{\boldsymbol{\theta}} \quad \mathbb{E}\big[\mathcal{L}(\boldsymbol{\theta}\,,\mathbf{Z})\big],$$

where the expectation is taken over $\mathbf{Z}$ and possible regularization terms are absorbed in the loss function $\mathcal{L}(\cdot)$. Notice that this formulation is quite general and can include regression, classification, clustering, or dimensionality reduction tasks as special cases. As an example, in the case of linear regression $\mathbf{Z} = (\mathbf{X}, Y)$ and $\mathcal{L}(\boldsymbol{\theta}\,,\mathbf{Z}) = (Y - \boldsymbol{\theta}^T\mathbf{X})^2$ where $\mathbf{X}$ is a random vector and $Y$ is the random target variable.

Assume that, in addition to minimizing the average loss, we are interested in bringing fairness to our learning task. Let $S$ be a sensitive attribute (such as age or gender) and $\widehat{Y}_{\boldsymbol{\theta}}(\mathbf{Z})$ be a certain output of our inference task using parameter $\boldsymbol{\theta}$. Assume we are interested in removing/reducing the dependence between the random variable $\widehat{Y}_{\boldsymbol{\theta}}(\mathbf{Z})$ and the sensitive attribute $S$. To balance the goodness-of-fit and fairness, one can solve the following optimization problem

$$\min_{\boldsymbol{\theta}} \quad \mathbb{E}\big[\mathcal{L}(\boldsymbol{\theta},\mathbf{Z})\big] + \lambda\,\rho_R^2\big(\widehat{Y}_{\boldsymbol{\theta}}(\mathbf{Z}), S\big), \tag{2}$$

where $\lambda$ is a positive scalar balancing fairness and goodness-of-fit. Notice that the above framework is quite general. For example, $\hat{Y}_\theta$ may be the assigned label in a classification task, the assigned cluster in a clustering task, or the output of a regressor in a regression task.

Using the definition of Rényi correlation, we can rewrite optimization problem in equation 2 as

$$\min_{\boldsymbol{\theta}} \sup_{f,g} \quad \mathbb{E}\big[\mathcal{L}(\boldsymbol{\theta},\mathbf{Z})\big] + \lambda\,\big(\mathbb{E}\big[f(\widehat{Y}_{\boldsymbol{\theta}}(\mathbf{Z}))\,g(S)\big]\big)^2,$$

$$\text{s.t.} \quad \mathbb{E}\big[f(\widehat{Y}_{\boldsymbol{\theta}}(\mathbf{Z}))\big] = \mathbb{E}\big[g(S)\big] = 0, \quad \mathbb{E}\big[f^2(\widehat{Y}_{\boldsymbol{\theta}}(\mathbf{Z}))\big] = \mathbb{E}\big[g^2(S)\big] = 1, \tag{3}$$

where the supremum is taken over the set of measurable functions. The next natural question to ask is whether this optimization problem can be efficiently solved in practice. This question motivates the discussions of the following subsection.

## 3.1 Computing Rényi Correlation

The objective function in equation 3 may be non-convex in $\boldsymbol{\theta}$ in general. Several algorithms have been recently proposed for solving such non-convex min-max optimization problems Sanjabi et al. (2018); Nouiehed et al. (2019); Jin et al. (2019). Most of these methods require solving the inner maximization problem to (approximate) global optimality. More precisely, we need to be able to solve the optimization problem described in equation 1. While popular heuristic approaches such as parameterizing the functions $f$ and $g$ with neural networks can be used to solve equation 1, we focus on solving this problem in a more rigorous manner. In particular, we narrow down our focus to the discrete random variable case. This case holds for many practical sensitive attributes among which are the gender and race. In what follows, we show that in this case, equation 1 can be solved "efficiently" to global optimality.

**Theorem 3.1** (Witsenhausen (1975)). *Let $a \in \{a_1, \ldots, a_c\}$ and $b \in \{b_1, \ldots, b_d\}$ be two discrete random variables. Then the Rényi coefficient $\rho_R(a,b)$ is equal to the second largest singular value of the matrix $\mathbf{Q} = [q_{ij}]_{i,j} \in \mathbb{R}^{c \times d}$, where $q_{ij} = \frac{\mathbb{P}(a=a_i, b=b_j)}{\sqrt{\mathbb{P}(a=a_i)\mathbb{P}(b=b_j)}}$.*

The above theorem provides a computationally tractable approach for computing the Rényi coefficient. This computation could be further simplified when one of the random variables is binary.

**Theorem 3.2.** *Suppose that $a \in \{1, \ldots, c\}$ is a discrete random variable and $b \in \{0, 1\}$ is a binary random variable. Let $\tilde{\mathbf{a}}$ be a one-hot encoding of $a$, i.e., $\tilde{\mathbf{a}} = \mathbf{e}_i$ if $a = i$, where $\mathbf{e}_i = (0, \ldots, 0, 1, 0 \ldots, 0)$ is the $i$-th standard unit vector. Let $\tilde{b} = b - 1/2$. Then,*

$$\rho_R(a, b) = \sqrt{1 - \frac{\gamma}{\mathbb{P}(b=1)\mathbb{P}(b=0)}},$$

*where $\gamma \triangleq \min_{\mathbf{w}\in\mathbb{R}^c} \quad \mathbb{E}\left[(\mathbf{w}^T\tilde{\mathbf{a}} - \tilde{b})^2\right]$. Equivalently,*

$$\gamma \triangleq \min_{\mathbf{w}\in\mathbb{R}^c} \quad \sum_{i=1}^c w_i^2 \mathbb{P}(a=i) - \sum_{i=1}^c w_i\big(\mathbb{P}(a=i, b=1) - \mathbb{P}(a=i, b=0)\big) + 1/4.$$

*Proof.* The proof is relegated to the appendix. □

Let us specialize our framework to classification and clustering problems in the next two sections.

## 4 RÉNYI FAIR CLASSIFICATION

In a typical (multi-class) classification problem, we are given samples from a random variable $\mathbf{Z} \triangleq (\mathbf{X}, Y)$ and the goal is to predict $Y$ from $\mathbf{X}$. Here $\mathbf{X} \in \mathbb{R}^d$ is the input feature vector, and $Y \in \mathcal{Y} \triangleq \{1, \ldots, c\}$ is the class label. Let $\widehat{Y}_{\boldsymbol{\theta}}$ be the output of our classifier taking different values in the set $\{1, \ldots, c\}$. Assume further that

$$\mathbb{P}(\widehat{Y}_{\boldsymbol{\theta}} = i \mid \mathbf{X}) = \mathcal{F}_i(\boldsymbol{\theta}, \mathbf{X}), \quad \forall i = 1, \ldots, c.$$

Here $\boldsymbol{\theta}$ is that parameter of the classifier that needs to be tuned. For example, $\mathcal{F}(\boldsymbol{\theta}, \mathbf{X}) = (\mathcal{F}_1(\boldsymbol{\theta}, \mathbf{X}), \ldots, \mathcal{F}_c(\boldsymbol{\theta}, \mathbf{X}))$ could represent the output of a neural network after *softmax* layer; the soft probability label assigned by a logistic regression model; or the 0-1 probability values obtained by a deterministic classifier. In order to find the optimal parameter $\boldsymbol{\theta}$, we need to solve the optimization problem

$$\min_{\boldsymbol{\theta}} \ \mathbb{E}\Big[\mathcal{L}(\mathcal{F}(\boldsymbol{\theta}, \mathbf{X}), Y)\Big], \tag{4}$$

where $\mathcal{L}$ is the loss function and the expectation is taken over the random variable $\mathbf{Z} = (\mathbf{X}, Y)$. Let $S$ be the sensitive attribute. We say a model satisfies *demographic parity* if the assigned label $\widehat{Y}$ is independent of the sensitive attribute $S$, see Dwork et al. (2012). Using our regularization framework, to find the optimal parameter $\boldsymbol{\theta}$ balancing classification accuracy and fairness objective, we need to solve

$$\min_{\boldsymbol{\theta}} \ \mathbb{E}\Big[\mathcal{L}(\mathcal{F}(\boldsymbol{\theta}, \mathbf{X}), Y)\Big] + \lambda \rho_R^2(\widehat{Y}_{\boldsymbol{\theta}}, S). \tag{5}$$

### 4.1 GENERAL DISCRETE CASE

When $S \in \{s_1, \ldots, s_d\}$ is discrete, Theorem 3.1 implies that equation 5 can be rewritten as

$$\min_{\boldsymbol{\theta}} \ \max_{\mathbf{v} \perp \mathbf{v}_1, \|\mathbf{v}\|^2 \leq 1} \left( f_D(\boldsymbol{\theta}, \mathbf{v}) \triangleq \mathbb{E}\Big[\mathcal{L}(\mathcal{F}(\boldsymbol{\theta}, \mathbf{X}), \mathbf{Y})\Big] + \lambda \mathbf{v}^T \mathbf{Q}_{\boldsymbol{\theta}}^T \mathbf{Q}_{\boldsymbol{\theta}} \mathbf{v} \right). \tag{6}$$

Here $\mathbf{v}_1 = \left[\sqrt{\mathbb{P}(S = s_1)}, \ldots, \sqrt{\mathbb{P}(S = s_d)}\right] \in \mathbb{R}^d$ is the right singular vector corresponding to the largest singular value of $\mathbf{Q}_{\boldsymbol{\theta}} = [q_{ij}]_{i,j} \in \mathbb{R}^{c \times d}$, with $q_{ij} \triangleq \dfrac{\mathbb{P}(\widehat{Y}_{\boldsymbol{\theta}} = i \mid S = s_j)\mathbb{P}(S = s_j)}{\sqrt{\mathbb{P}(\widehat{Y}_{\boldsymbol{\theta}} = i)\,\mathbb{P}(S = s_j)}}$.

Given training data $(\mathbf{x}_n, y_n)_{n=1}^N$ sampled from the random variable $\mathbf{Z} = (\mathbf{X}, Y)$, we can estimate the entries of the matrix $\mathbf{Q}_{\boldsymbol{\theta}}$ using $\mathbb{P}(\widehat{Y}_{\boldsymbol{\theta}} = i) = \mathbb{E}[\mathbb{P}(\widehat{Y}_{\boldsymbol{\theta}} = i \mid \mathbf{X})] \approx \frac{1}{N} \sum_{n=1}^N \mathcal{F}_i(\boldsymbol{\theta}, \mathbf{x}_n)$, and $\mathbb{P}(\widehat{Y}_{\boldsymbol{\theta}} = i \mid S = s_j) \approx \frac{1}{|\mathcal{X}_j|} \sum_{\mathbf{x} \in \mathcal{X}_j} \mathcal{F}_i(\boldsymbol{\theta}, \mathbf{x})$, where $\mathcal{X}_j$ is the set of samples with sensitive attribute $s_j$. Motivated by the algorithm proposed in Jin et al. (2019), we present Algorithm 1 for solving equation 6.

---

**Algorithm 1** Rényi Fair Classifier for Discrete Sensitive Attributes

---
1: **Input**: $\boldsymbol{\theta}^0 \in \Theta$, step-size $\eta$.
2: **for** $t = 0, 1, \ldots, T$ **do**
3:     Set $\mathbf{v}^{t+1} \leftarrow \max_{\mathbf{v} \in \perp \mathbf{v}_1, \|\mathbf{v}\| \leq 1} f_D(\boldsymbol{\theta}^t, \mathbf{v})$ by finding the second singular vector of $\mathbf{Q}_{\boldsymbol{\theta}^t}$
4:     Set $\boldsymbol{\theta}^{t+1} \leftarrow \boldsymbol{\theta}^t - \eta \nabla_{\boldsymbol{\theta}} f_D(\boldsymbol{\theta}^t, \mathbf{v}^{t+1})$
5: **end for**

---

To understand the convergence behavior of Algorithm 1 for the nonconvex optimization problem equation 6, we need to first define an approximate stationary solution. Let us define $g(\boldsymbol{\theta}) =$

$\max_{\mathbf{v} \in \perp \mathbf{v}_1, \|\mathbf{v}\| \leq 1} f(\boldsymbol{\theta}, \mathbf{v})$. Assume further that $f(\cdot, \mathbf{v})$ has $L_1$-Lipschitz gradient, then $g(\cdot)$ is $L_1$-weakly convex; for more details check Rafique et al. (2018). For such weakly convex function, we say $\boldsymbol{\theta}^*$ is a $\epsilon$-stationary solution if the gradient of its Moreau envelop is smaller than epsilon, i.e., $\|\nabla g_\beta(\cdot)\| \leq \varepsilon$ with $g_\beta(\boldsymbol{\theta}) \triangleq \min_{\boldsymbol{\theta}'} g(\boldsymbol{\theta}') + \frac{1}{2\beta} \|\boldsymbol{\theta} - \boldsymbol{\theta}'\|$ and $\beta < \frac{1}{2L_1}$ is a given constant. The following theorem demonstrates the convergence of Algorithm 1. This theorem is a direct consequence of Theorem 27 in Jin et al. (2019).

**Theorem 4.1.** *Suppose that $f$ is $L_0$-Lipschitz and $L_1$-gradient Lipschitz. Then Algorithm 1 computes an $\varepsilon$-stationary solution of the objective function in equation 6 in $\mathcal{O}(\varepsilon^{-4})$ iterations.*

## 4.2 BINARY CASE

When $S$ is binary, we can obtain a more efficient algorithm compared to Algorithm 1 by exploiting Theorem 3.2. Particularly, by a simple scaling of $\lambda$ and ignoring the constant terms, the optimization problem in equation 5 can be written as

$$\min_{\boldsymbol{\theta}} \max_{\mathbf{w}} \quad f(\boldsymbol{\theta}, \mathbf{v}) \triangleq \mathbb{E}\Big[\mathcal{L}\big(\mathcal{F}(\boldsymbol{\theta}, \mathbf{X}), Y\big)\Big] - \lambda\Big[\sum_{i=1}^c w_i^2 \mathbb{P}\big(\widehat{Y}_{\boldsymbol{\theta}} = i\big) - \sum_{i=1}^c w_i\big(\mathbb{P}\big(\widehat{Y}_{\boldsymbol{\theta}} = i, S = 1\big) - \mathbb{P}\big(\widehat{Y}_{\boldsymbol{\theta}} = i, S = 0\big)\big)\Big]. \tag{7}$$

Defining $\tilde{S} = 2S - 1$, the above problem can be rewritten as

$$\min_{\boldsymbol{\theta}} \max_{\mathbf{w}} \quad \mathbb{E}\Big[\mathcal{L}(\mathcal{F}(\boldsymbol{\theta}, \mathbf{X}), Y) - \lambda \sum_{i=1}^c w_i^2 \mathcal{F}_i(\boldsymbol{\theta}, \mathbf{X}) + \lambda \sum_{i=1}^c w_i \widetilde{S} \mathcal{F}_i(\boldsymbol{\theta}, \mathbf{X})\Big].$$

Thus, given training data $(\mathbf{x}_n, y_n)_{n=1}^N$ sampled from the random variable $\mathbf{Z} = (\mathbf{X}, Y)$, we solve

$$\min_{\boldsymbol{\theta}} \max_{\mathbf{w}} \left[ f_B(\boldsymbol{\theta}, \mathbf{w}) \triangleq \frac{1}{N} \sum_{n=1}^N \Big[\mathcal{L}\big(\mathcal{F}(\boldsymbol{\theta}, \mathbf{x}_n), y_n\big) - \lambda \sum_{i=1}^c w_i^2 \mathcal{F}_i(\boldsymbol{\theta}, \mathbf{x}_n) + \lambda \sum_{i=1}^c w_i \widetilde{s}_n \mathcal{F}_i(\boldsymbol{\theta}, \mathbf{x}_n)\Big] \right]. \tag{8}$$

Notice that the maximization problem in equation 8 is concave, separable, and has a closed-form solution. We propose Algorithm 2 for solving equation 8.

---

**Algorithm 2** Rényi Fair Classifier for Binary Sensitive Attributes

1: **Input**: $\boldsymbol{\theta}^0 \in \Theta$, step-size $\eta$.
2: **for** $t = 0, 1, \ldots, T$ **do**
3:     Set $\mathbf{w}^{t+1} \leftarrow \arg\max_{\mathbf{w}} f_B(\boldsymbol{\theta}^t, \mathbf{w})$, i.e., set $w_i^{t+1} \leftarrow \frac{\sum_{n=1}^N \widetilde{s}_n \mathcal{F}_i(\boldsymbol{\theta}^t, \mathbf{x}_n)}{2 \sum_{n=1}^N \mathcal{F}_i(\boldsymbol{\theta}^t, \mathbf{x}_n)}, \quad \forall i = 1, \ldots, c$
4:     Set $\boldsymbol{\theta}^{t+1} \leftarrow \boldsymbol{\theta}^t - \eta \nabla_{\boldsymbol{\theta}} f_B(\boldsymbol{\theta}^t, \mathbf{w}^{t+1})$
5: **end for**

---

While the result in Theorem 4.1 can be applied to Algorithm 2, under the following assumption, we can show a superior convergence rate.

**Assumption 4.1.** *We assume that there exists a constant scalar $\mu > 0$ such that $\sum_{n=1}^N \mathcal{F}_i(\boldsymbol{\theta}, \mathbf{x}_n) \geq \mu, \quad \forall i = 1, \ldots, C$.*

This assumption is reasonable when soft-max is used. This is because we can always assume $\boldsymbol{\theta}$ lies in a compact set in practice, and hence the output of the softmax layer cannot be arbitrarily small.

**Theorem 4.2.** *Suppose that $f$ is $L_1$-gradient Lipschitz. Then Algorithm 2 computes an $\varepsilon$-stationary solution of the objective function in equation 8 in $\mathcal{O}(\varepsilon^{-2})$ iterations.*

*Proof.* The proof is relegated to the appendix. ☐

Notice that this convergence rate is clearly a faster rate than the one obtained in Theorem 4.1. Moreover, this rate of convergence matches the oracle lower bound for general non-convex optimization;

see Carmon et al. (2019). This observation shows that the computational overhead of imposing fairness is negligible as compared to solving the original non-convex training problem without imposing fairness.

**Remark 4.3** (Extension to multiple sensitive attributes). *Our discrete Rényi classification framework can naturally be extended to the case of multiple discrete sensitivity attributes by concatenating all attributes into one. For instance, when we have two sensitivity attribute $S^1 \in \{0, 1\}$ and $S^2 \in \{0, 1\}$, we can consider them as a single attribute $S \in \{0, 1, 2, 3\}$ corresponding to the four combinations of $\{(S^1 = 0, S^2 = 0), (S^1 = 0, S^2 = 1), (S^1 = 0, S^2 = 0), (S^1 = 1, S^2 = 1)\}$.*

**Remark 4.4** (Extension to other notions of fairness). *Our proposed framework imposes the demographic parity notion of group fairness. However, other notions of group fairness may be represented by (conditional) independence conditions. For such cases, we can again apply our framework. For example, we say a predictor $\hat{Y}_{\boldsymbol{\theta}}$ satisfies equalized odds condition if the predictor $\hat{Y}_{\boldsymbol{\theta}}$ is conditionally independent of the sensitive attribute $S$ given the true label $Y$. Similar to formulation equation 5, the equalized odds fairness notion can be achieved by the following min-max problem*

$$\min_{\boldsymbol{\theta}} \quad \mathbb{E}\left[\mathcal{L}\big(\mathcal{F}(\boldsymbol{\theta}, \mathbf{X}), Y\big)\right] + \lambda \sum_{y \in \mathcal{Y}} \rho_R^2\left(\hat{Y}_{\boldsymbol{\theta}}, S \mid Y = y\right). \tag{9}$$

## 5 RÉNYI FAIR CLUSTERING

In this section, we apply the proposed fair Rényi framework to the widespread $K$-means clustering problem. Given a set of data points $\mathbf{x}_1, \ldots, \mathbf{x}_N \in \mathbb{R}^{N \times d}$, in the $K$-means problem, we seek to partition them into $K$ clusters such that the following objective function is minimized:

$$\min_{\mathbf{A}, \mathbf{C}} \sum_{n=1}^{N} \sum_{k=1}^{K} a_{kn} \|\mathbf{x}_n - \mathbf{c}_k\|^2 \quad \text{s.t.} \quad \sum_{k=1}^{K} a_{kn} = 1, \; \forall n, \quad a_{kn} \in \{0, 1\}, \; \forall k, n \tag{10}$$

where $\mathbf{c}_k$ is the centroid of cluster $k$; the variable $a_{kn} = 1$ if data point $\mathbf{x}_n$ belongs to cluster $k$ and it is zero otherwise; $\mathbf{A} = [a_{kn}]_{k,n}$ and $\mathbf{C} = [\mathbf{c}_1, \ldots, \mathbf{c}_K]$ represent the association matrix and the cluster centroids respectively. Now, suppose we have an additional sensitive attribute $S$ for each one of the given data points. In order to have a fair clustering under disparate impact doctrine, we need to make the random variable $\mathbf{a}_n = [a_{1n}, \ldots, a_{Kn}]$ independent of $S$. In other words, we need to make the clustering assignment independent of the sensitive attribute $S$. Using our framework in equation 2, we can easily add a regularizer to this problem to impose fairness under disparate impact doctrine. In particular, for binary sensitive attribute $S$, using Theorem 3.2, and absorbing the constants into the hyper-parameter $\lambda$, we need to solve

$$\min_{\mathbf{A}, \mathbf{C}} \max_{\mathbf{w} \in \mathbb{R}^K} \sum_{n=1}^{N} \sum_{k=1}^{K} a_{kn} \|\mathbf{x}_n - \mathbf{c}_k\|^2 - \lambda \sum_{n=1}^{N} (\mathbf{a}_n^T \mathbf{w} - s_n)^2 \tag{11}$$
$$\text{s.t.} \quad \sum_{k=1}^{K} a_{kn} = 1, \quad \forall n, \quad a_{kn} \in \{0, 1\}, \quad \forall k, n.$$

where $\mathbf{a}_n = (a_{1n}, \ldots, a_{Kn})^T$ encodes the clustering information of data point $\mathbf{x}_n$ and $s_n$ is the sensitive attribute for data point $n$.

Fixing the assignment matrix $\mathbf{A}$, and cluster centers $\mathbf{C}$, the vector $\mathbf{w}$ can be updated in closed-form. More specifically, $w_k$ at each iteration equals to the current proportion of the privileged group in the $k$-th cluster. Combining this idea with the update rules of assignments and cluster centers in the standard K-means algorithm, we propose Algorithm 3, which is a fair $K$-means algorithm under disparate impact doctrine. To illustrate the behavior of the algorithm, a toy example is presented in Appendix C.

The main difference between this algorithm and the popular $K$-means algorithm is in Step 6 of Algorithm 3. This step is a result of optimizing equation 11 over $\mathbf{A}$ when both $\mathbf{C}$ and $\mathbf{w}$ are fixed. When $\lambda = 0$, this step would be identical to the update of cluster assignment variables in $K$-means. However, when $\lambda > 0$, Step 6 considers fairness when computing the distance considered in updating the cluster assignments.

---

**Algorithm 3** Rényi Fair K-means

---

1: **Input**: $\mathbf{X} = \{\mathbf{x}_1, \ldots, \mathbf{x}_N\}$ and $\mathbf{S} = \{s_1, \ldots, s_N\}$
2: **Initialize**: Random assignment $\mathbf{A}$ s.t. $\sum_{k=1}^{K} a_{kn} = 1 \, \forall n$; and $a_{kn} \in \{0, 1\}$. *Set* $\mathbf{A}_{prev} = \mathbf{0}$.
3: **while** $\mathbf{A}_{prev} \neq \mathbf{A}$ **do**
4:      Set $\mathbf{A}_{prev} = \mathbf{A}$
5:      **for** $n = 1, \ldots, N$ **do**                                       ▷ Update $\mathbf{A}$
6:          $k^* = \arg\min_k \|\mathbf{x}_n - \mathbf{c}_k\|^2 - \lambda(\mathbf{w}_k - s_n)^2$
7:          Set $a_{k^*n} = 1$ and $a_{kn} = 0$ for all $k \neq k^*$
8:          Set $w_k = \dfrac{\sum_{n=1}^{N} s_n a_{kn}}{\sum_{n=1}^{N} a_{kn}}, \; \forall k = 1, \ldots, K.$              ▷ Update $\mathbf{w}$
9:      **end for**
10:     Set $\mathbf{c}_k = \dfrac{\sum_{n=1}^{N} a_{kn} \mathbf{x}_n}{\sum_{n=1}^{N} a_{kn}}, \; \forall k = 1, \ldots, K.$                ▷ Update $\mathbf{c}$
11: **end while**

---

**Remark 5.1.** *Note that in Algorithm 3, the parameter* $\mathbf{w}$ *is being updated after each assignment of a point to a cluster. More specifically, for every iteration of the algorithm,* $\mathbf{w}$ *is updated* $N$ *times. If we otherwise update* $\mathbf{w}$ *after completely updating the matrix* $\mathbf{A}$*, then with a simple counterexample we can show that the algorithm can get stuck; see more details in Appendix C.1.*

## 6   NUMERICAL EXPERIMENTS

In this section, we evaluate the performance of the proposed Rényi fair classifier and Rényi fair k-means algorithm on three standard datasets: *Bank*, *German Credit*, and *Adult* datasets. The detailed description of these datasets is available in the supplementary material.

We evaluate the performance of our proposed Rényi classifier under both demographic parity, and equality of opportunity notions. We have implemented a logistic regression classifier regularized by Rényi correlation on Adult dataset considering gender as the sensitive feature. To measure the equality of opportunity we use the Equality of Opportunity (EO) violation, defined as EO Violation $= \left| \mathbb{P}(\hat{Y} = 1|S = 1, Y = 1) - \mathbb{P}(\hat{Y} = 1|S = 0, Y = 1) \right|$, where $\hat{Y}$ and $Y$ represent the predicted, and true labels respectively. Smaller EO violation corresponds to a more fair solution. Figure 1, parts (a) and (b) demonstrate that by increasing $\lambda$, the Rényi regularizer coefficient decreases implying a more fair classifier at the price of a higher training and testing errors. Figure 1, part (c) compares the fair Rényi logistic regression model to several existing methods in the literature Hardt et al. (2016); Zafar et al. (2015); Rezaei et al. (2019); Donini et al. (2018). As we can see in plot (c), the Rényi classifier outperforms other methods in terms of accuracy for a given level of EO violation.

The better performance of the Rényi fair classifier compared to the baselines could be attributed to the following. Hardt et al. (2016) is a post-processing approach where the output of the classification is modified to promote a fair prediction. This modification is done without changing the classification process. Clearly, this approach limits the design space and cannot explore the possibilities that can reach with "in-processing" methods where both fairness and classification objectives are optimized jointly. Zafar et al. (2015) imposes fairness by using linear covariance and thus can only capture linear dependence between the predictor and the sensitive attribute. Consequently, there might exist nonlinear dependencies which are revealed in fairness measures such as DP or EO violation. Rezaei et al. (2019); Donini et al. (2018), on the other hand, propose to use nonlinear measures of dependence as regularizers. However, due to computational barriers, they approximate the regularizer and solve the approximate problem. The approximation step could potentially have an adverse effect on the performance of the resulting classifier. Notice that while these methods are different in terms of the way they impose fairness, they are all implemented for logistic regression model (with the exception of SVM model used in Donini et al. (2018)). Thus, the difference in the performance is not due to the classification model used in the experiments.

To show the practical benefits of Rényi correlation over Pearson correlation and HSIC regularizers under the demographic parity notion, we evaluate the logistic regression classifier regularized

by these three measures on Adult, Bank, and German Credit datasets. For the first two plots, we use $p\% = \min(\frac{\mathbb{P}(\hat{Y}=1|S=1)}{\mathbb{P}(\hat{Y}=1|S=0)}, \frac{\mathbb{P}(\hat{Y}=1|S=0)}{\mathbb{P}(\hat{Y}=1|S=1)})$ as a measure of fairness. Since $p\%$ is defined only for binary sensitive variables, for the last two plots in Figure 2 (German dataset with gender and marital status, and Adult dataset with gender and race as the sensitive features), we use the inverse of demographic parity (DP) violation as the fairness measure. We define DP violation as DP Violation $= \max_{a,b}|\mathbb{P}(\hat{Y}=1|S=a) - \mathbb{P}(\hat{Y}=1|S=b)|$. As it is evident from the figure, Rényi classifier outperforms both HSIC and Pearson classifiers, especially when targeting high levels of fairness. For the last two experiments in Figure 2, we could not further increase fairness by increasing the regularization coefficient for Pearson and HSIC regularizers (see green and red curves cannot go beyond a certain point on the fairness axis). This can be explained by the nonlinear correlation between the predictor and the sensitive variables in these two scenarios which cannot be fully captured using linear or quadratic independence measures. Interestingly, our experiments indicate that minimizing Rényi correlation eventually minimizes the Normalized Mutual Information (NMI) between the variables (See Supplementary Figure 6). Recall that similar to Rényi correlation, NMI can capture any dependence between two given random variables.

Finally, to evaluate the performance of our fair k-means algorithm, we implement Algorithm 3 to find clusters of Adult and Bank datasets. We use the deviation of the elements of the vector $\mathbf{w}$ as a measure of fairness. The element $\mathbf{w}_k$ of $\mathbf{w}$ represents the ratio of the number of data points that belong to the privileged group ($S = 1$) in cluster $k$ over the number of data points in that cluster. This notion of fairness is closely related to *minimum balance* introduced by Chierichetti et al. (2017). The deviation of these elements is a measure for the deviation of these ratios across different clusters. A clustering solution is exactly fair if all entries of $\mathbf{w}$ are the same. For $K = 14$, we plot in Figure 3 the minimum, maximum, average, and average $\pm$ standard deviation of the entries of $\mathbf{w}$ vector for different values of $\lambda$. For an exactly fair clustering solution, these values should be the same. As we can see in Figure 3, increasing $\lambda$ yields exact fair clustering at the price of a higher clustering loss.

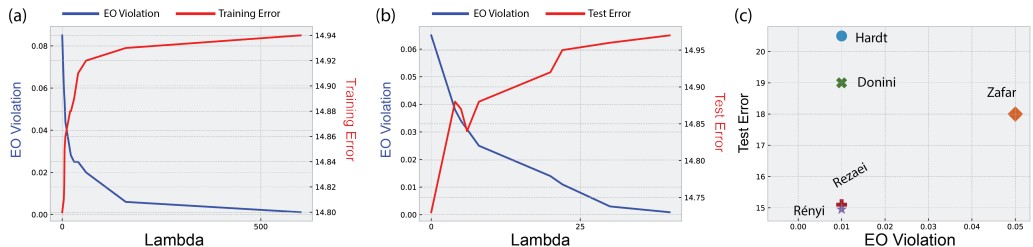

Figure 1: Trade-off between the accuracy of classifier and fairness on the adult dataset under the equality of opportunity notion. (a, b) By increasing $\lambda$ from 0 to 1000, EO violation (the blue curve on the left axis) approaches to 0. The fairer solution comes at the price of a slight increase of the training/test error (Red curve, right axis). (c) Comparison of the existing approaches with Rényi classifier, under the equality of opportunity notion. Rényi classifier demonstrates a better accuracy for a given level of fairness measured by EO violation.

## 7 CONCLUSION

In this paper, we proposed Rényi fair inference as an in-process method to impose fairness in empirical risk minimization. Fairness is defined as (conditional) independence between a sensitive attribute and the inference output from the learning machine. As statistical independence is only measurable when the data distributions are fully known, we can only hope to promote independence through empirical surrogates in this framework. Our method imposes a regularizer in the form of the Rényi correlation (maximal correlation) between a sensitive attribute(s) and the inference output. Rényi correlation between two random variables is zero if and only if they are independent, which is a desirable property for an independence surrogate. We pose Rényi fair correlation as a minimax optimization problem. In the case where the sensitive attributes are discrete (e.g., race), we present an algorithm that finds a first-order optimal solution to the problem with convergence guarantees. In the special case where the sensitive attribute is binary (e.g., gender), we show an algorithm with optimal convergence guarantees. Our numerical experiments show that Rényi fair inference captures

nonlinear correlations better than Pearson correlation or HSIC. We also show that increasing the regularization hyperparameter results in near statistical independence between the sensitive attribute and the inference output. Future work would naturally consider extension to continuous sensitive attributes and problems with missing or non-explicit sensitive labels such as fair word embedding problem.

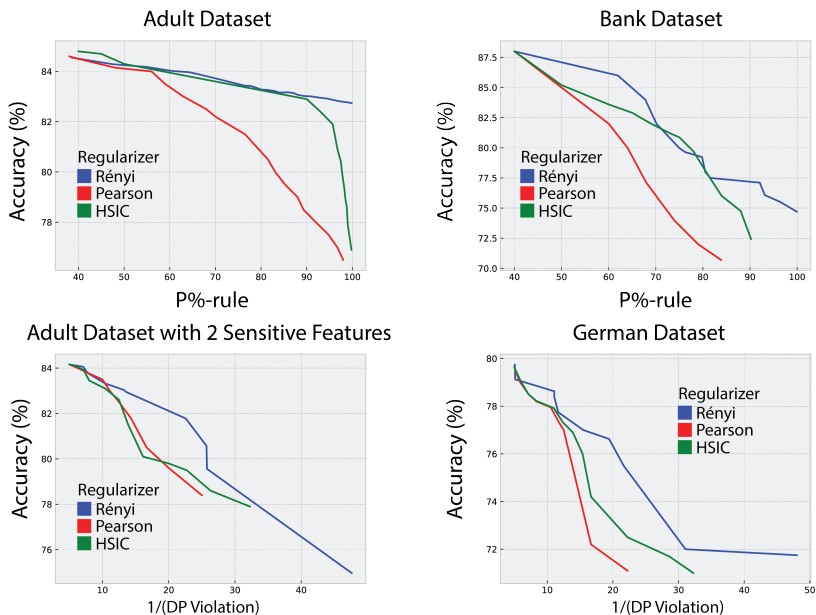

Figure 2: Trade-off between accuracy and fairness for logistic regression classifier regularized with Rényi, HSIC, and Pearson measures, on German Credit, Adult, and Bank datasets. (Top) The drop in the accuracy of the model regularized by Rényi, is less than the same model regularized by HSIC, and Pearson correlation. Moreover, as can be observed for both Bank and Adult datasets, Pearson and HSIC regularizers usually cannot increase $p\%$ beyond a certain limit, due to the fact that removing all linear correlations does not guarantee independence between the predictor and the sensitive attribute. (Down) When the sensitive attribute is not binary (or we have more than one sensitive attribute), obtaining a fair model for HSIC and Pearson regularizers is even harder. The model regularized by HSIC or Pearson, cannot minimize the DP violation (or maximize its reciprocal) beyond a threshold.

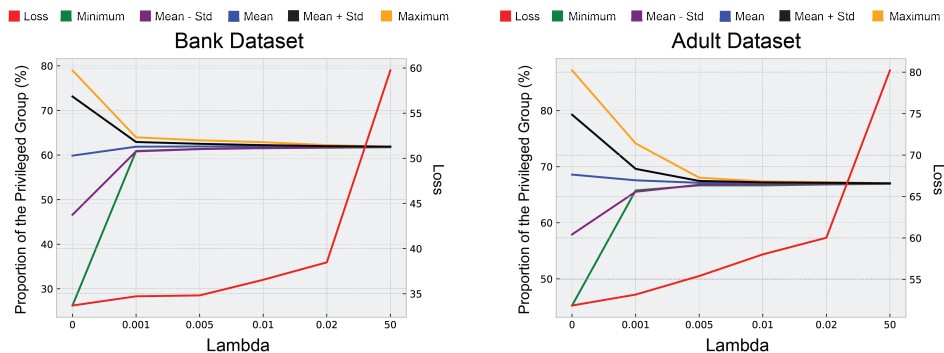

Figure 3: Performance and fairness of $K$-means algorithm in terms of Rényi regularizer hyper-parameter $\lambda$. By increasing $\lambda$, the standard deviation of the $w$ vector components (each component represents the relative proportion of the privileged group in the corresponding cluster) is reduced accordingly. Both plots demonstrate that the standard deviation of $w$ is reduced fast with respect to $\lambda$, and the increase in loss is small when $\lambda \leq 0.005$. However, to reach a completely fair clustering, a $\lambda \geq 1$ must be chosen that can increase the loss (the right axis, red curve) drastically.

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

# A    APPENDIX: PROOF OF THEOREM 3.2

*Proof.* First, notice that since $\tilde{\mathbf{a}}$ is a one-hot encoding of $a$, any function $f : \{1, \ldots, c\} \mapsto \mathbb{R}$ can be equivalently represented as $f(\tilde{\mathbf{a}}) = \mathbf{u}^T \tilde{\mathbf{a}}$ for some $\mathbf{u} \in \mathbb{R}^c$. Therefore, following the definition of Rényi correlation, we can write

$$
\begin{aligned}
\rho_R(a, b) = \max_{\mathbf{u}, g} \quad & \mathbb{E}\left[(\mathbf{u}^T \tilde{\mathbf{a}}) g(b)\right] \\
\text{s.t.} \quad & \mathbb{E}\left[(\mathbf{u}^T \tilde{\mathbf{a}})^2\right] \leq 1, \quad \mathbb{E}[\mathbf{u}^T \tilde{\mathbf{a}}] = 0 \\
& \mathbb{E}[g^2(b)] \leq 1, \quad \mathbb{E}[g(b)] = 0
\end{aligned} \quad .
$$

Notice that since $b$ is binary, there is a unique function $g(b) = \frac{b-q}{\sqrt{q(1-q)}}$ satisfying the constraints where $q \triangleq \mathbb{P}(b = 1)$. Therefore, the above optimization problem can be written as

$$
\begin{aligned}
\rho_R(a, b) = \max_{\mathbf{u}} \quad & \mathbf{u}^T \mathbb{E}\left[\tilde{\mathbf{a}} g(b)\right] \\
\text{s.t.} \quad & \mathbf{u}^T \mathbb{E}\left[\tilde{\mathbf{a}} \tilde{\mathbf{a}}^T\right] \mathbf{u} \leq 1 \\
& \mathbf{u}^T \mathbb{E}[\tilde{\mathbf{a}}] = 0
\end{aligned} \quad .
$$

The last constraint simply implies that $\mathbf{u}$ should be orthogonal to $\mathbf{p} \triangleq \mathbb{E}[\tilde{\mathbf{a}}]$, which is a stochastic vector capturing the distribution of $a$. Equivalently, we can write $\mathbf{u} = \left(\mathbf{I} - \frac{\mathbf{p}\mathbf{p}^T}{\|\mathbf{p}\|^2}\right) \mathbf{v}$ for some $\mathbf{v} \in \mathbb{R}^c$. Thus, we can simplify the above optimization problem as

$$
\begin{aligned}
\rho_R(a, b) = \max_{\mathbf{v}} \quad & \mathbf{v}^T \left(\mathbf{I} - \frac{\mathbf{p}\mathbf{p}^T}{\|\mathbf{p}\|^2}\right) \mathbb{E}\left[\tilde{\mathbf{a}} g(b)\right] \\
\text{s.t.} \quad & \mathbf{v}^T \left(\mathbf{I} - \frac{\mathbf{p}\mathbf{p}^T}{\|\mathbf{p}\|^2}\right) \operatorname{diag}(\mathbf{p}) \left(\mathbf{I} - \frac{\mathbf{p}\mathbf{p}^T}{\|\mathbf{p}\|^2}\right) \mathbf{v} \leq 1,
\end{aligned}
$$

where in the constraint, we used the equality $\mathbb{E}\left[\tilde{\mathbf{a}} \tilde{\mathbf{a}}^T\right] = \operatorname{diag}(\mathbf{p})$, which follows the definition. Let us do the change of variable $\hat{\mathbf{v}} = \operatorname{diag}(\sqrt{\mathbf{p}})(\mathbf{I} - \frac{\mathbf{p}\mathbf{p}^T}{\|\mathbf{p}\|^2})\mathbf{v}$. Then the above optimization can be simplified to

$$
\begin{aligned}
\rho_R(a, b) = \max_{\hat{\mathbf{v}}} \quad & \hat{\mathbf{v}}^T \operatorname{diag}(1/\sqrt{\mathbf{p}}) \mathbb{E}\left[\tilde{\mathbf{a}} g(b)\right] \\
\text{s.t.} \quad & \|\hat{\mathbf{v}}\| \leq 1.
\end{aligned}
$$

Clearly, this leads to

$$
\begin{aligned}
\rho_R^2(a, b) &= \left\|\operatorname{diag}\left(\frac{1}{\sqrt{\mathbf{p}}}\right) \mathbb{E}\left[\tilde{\mathbf{a}} g(b)\right]\right\|^2 \\
&= \sum_{i=1}^{c} \frac{1}{\mathbb{P}(a=i)} \left(\mathbb{P}(a=i, b=1)\sqrt{\frac{1-q}{q}} - \mathbb{P}(a=i, b=0)\sqrt{\frac{q}{1-q}}\right)^2,
\end{aligned} \tag{12}
$$

where in the last equality we use the fact that $g(1) = \sqrt{\frac{1-q}{q}}$ and $g(0) = -\sqrt{\frac{q}{1-q}}$. Define $p_{i0} \triangleq \mathbb{P}(a = i, b = 0)$ and $p_{i1} \triangleq \mathbb{P}(a = i, b = 0)$, $p_i \triangleq \mathbb{P}(a = i) = p_{i0} + p_{i1}$. Then, using simple

algebraic manipulations, we have that

$$
\begin{aligned}
\rho_R^2(a,b) &= \sum_{i=1}^{c} \frac{1}{p_i} \left( p_{i1}\sqrt{\frac{1-q}{q}} - p_{i0}\sqrt{\frac{q}{1-q}} \right)^2 \\
&= \sum_{i=1}^{c} \frac{(2p_{i1}(1-q) - 2p_{i0}q)^2}{4p_i q(1-q)} - \sum_{i=1}^{c} \frac{(p_{i0} - p_{i1})^2}{4p_i q(1-q)} + \sum_{i=1}^{c} \frac{(p_{i0} - p_{i1})^2}{4p_i q(1-q)} \\
&= \sum_{i=1}^{c} \frac{((3-2q)p_{i1} - (1+2q)p_{i0})((1-2q)p_{i1} + (1-2q)p_{i0})}{4p_i q(1-q)} + \sum_{i=1}^{c} \frac{(p_{i0} - p_{i1})^2}{4p_i q(1-q)} \\
&= \frac{1-2q}{4q(1-q)}((3-2q)q - (1+2q)(1-q)) + \sum_{i=1}^{c} \frac{(p_{i0} - p_{i1})^2}{4p_i q(1-q)} \\
&= 1 - \frac{1 - \sum_{i=1}^{c}(p_{i0} - p_{i1})^2/p_i}{4q(1-q)} = 1 - \frac{\gamma}{q(1-q)},
\end{aligned}
$$

where in the last equality we used the definition of $\gamma$ and the optimal value of equation 3.2. $\qquad\square$

# B   PROOF OF THEOREM 4.2

*Proof.* Define $g_B(\boldsymbol{\theta}) = \max_{\mathbf{w}} f_B(\boldsymbol{\theta}, \mathbf{w})$. Since the optimization problem $\max_{\mathbf{w}} f_B(\boldsymbol{\theta}, \mathbf{w})$ is strongly concave in $\mathbf{w}$, using Danskin's theorem (see Danskin (1967) and Bertsekas (1971)), we conclude that the function $g_B(\cdot)$ is differentiable. Moreover,

$$
\nabla_{\boldsymbol{\theta}} g_B(\bar{\boldsymbol{\theta}}) = \nabla_{\boldsymbol{\theta}} f_B(\bar{\boldsymbol{\theta}}, \bar{\mathbf{w}})
$$

where $\bar{\mathbf{w}} = \arg\max_{\mathbf{w}} f_B(\bar{\boldsymbol{\theta}}, \mathbf{w})$. Thus Algorithm 2 is in fact equivalent to the gradient descent algorithm applied to $g_B(\boldsymbol{\theta})$. Thus according to (Nesterov, 2018, Chapter 1), the algorithm finds a point with $\|\nabla g_B(\boldsymbol{\theta})\| \leq \epsilon$ in $\mathcal{O}(\epsilon^{-2})$ iterations. $\qquad\square$

# C   RÉNYI FAIR K-MEANS

To illustrate the behavior of Algorithm 3, we deployed a simple two-dimensional toy example. In this example we generated data by randomly selecting 5 center points and then randomly generating 500 data points around each center according to a normal distribution with small enough variance. The data is shown in Figure 4 with different colors corresponding to different clusters. Moreover, we assigned for each data point $x_i$ a binary value $s_i \in \{0,1\}$ that corresponds to its sensitive attribute. This assignment was also performed randomly except for points generated around center 2 (green points in Figure 4) which were assigned a value of 1 and points generated around center 4 (blue points in Figure 4) which were assigned a value of 0. Without imposing fairness, traditional K-means algorithm would group points generated around center 2 in one cluster regardless of the fact that they all belong to the same protected group. Similarly, points generated around center 4 will belong to the same cluster. Hence, according to traditional K-means clustering shown in Figure 4, the proportion of the protected group in clusters 2 and 4 are 1 and 0 respectively. However, when imposing our fairness scheme, we expect these points to be distributed among various clusters to achieve balanced clustering. This is illustrated in Figure 5. It is evident from Figure 5 that increasing lambda, data points corresponding to centers 2 and 4 are now distributed among different clusters.

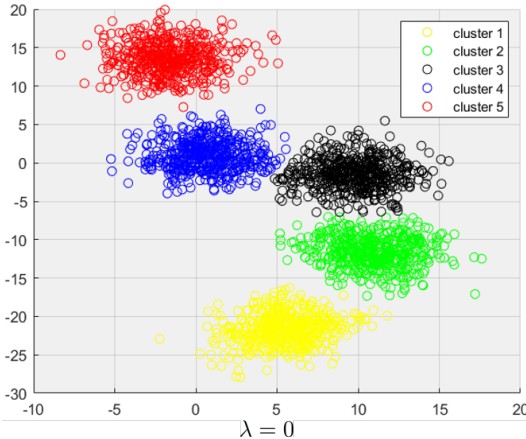

Figure 4: Applying K-means algorithm without fairness on the synthetic dataset.

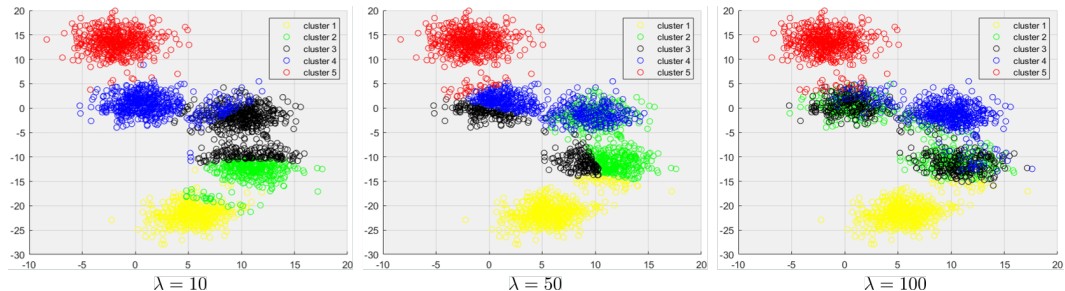

Figure 5: Applying fair K-means algorithm with different values of $\lambda$ on the synthetic dataset.

### C.1 Updating $\mathbf{w}$ after updating the assignment of each data point in Algorithm 3

To understand the reasoning behind updating the vector of proportions $\mathbf{w}$ after updating each $\mathbf{a}_i$ which is the assignment of data point $i$, we discuss a simple one-dimensional counterexample. Consider the following four data points $X_1 = -5$, $X_2 = -4$, $X_3 = 4$, and $X_4 = 5$ with their corresponding sensitive attributes $S_1 = S_2 = 1$ and $S_3 = S_4 = 0$. Moreover, assume the following initial $\mathbf{A}^0$ and $\mathbf{C}^0$

$$\mathbf{A}^0 = \begin{bmatrix} 1 & 0 \\ 1 & 0 \\ 0 & 1 \\ 0 & 1 \end{bmatrix}, \quad \mathbf{C}^0 = [-4.5, 4.5].$$

Hence, $X_1$ and $X_2$ which both have a sensitive attribute of 1 are assigned to cluster 1 with center $\mathbf{C}_1^0 = -4.5$ and $X_3$ and $X_4$ which both have a sensitive attribute of 0 are assigned to cluster 2 with center $\mathbf{C}_2^0 = 4.5$. Then $\mathbf{w}$ which is the current proportion of the privileged group in the clusters will be $\mathbf{w}^0 = [1, 0]$. Now, for sufficiently large $\lambda$ if we update $\mathbf{A}$ according to Step 6 of Algorithm 3, we get the following new assignment

$$\mathbf{A}^1 = \begin{bmatrix} 0 & 1 \\ 0 & 1 \\ 1 & 0 \\ 1 & 0 \end{bmatrix}, \quad \mathbf{C}^1 = [4.5, -4.5], \quad \mathbf{w}^1 = [0, 1].$$

Hence, the points just switch their clusters. Then, performing another iteration will get us back to the initial setup and the algorithm will get stuck between these two states that are both not fair. To overcome this issue we update the proportions $\mathbf{w}$ after updating the assignment of each data point.

## D    DATASETS DESCRIPTION

In this section we introduce the datasets used in numerical experiment discussed in Section 6. All of these datasets are publicly available at UCI repository.

- **German Credit Dataset**:[1] German Credit dataset consists of 20 features (13 categorical and 7 numerical) regarding to social, and economic status of 1000 customers. The assigned task is to classify customers as good or bad credit risks. Without imposing fairness, the DP violation of the trained model is larger than 20%. We chose first 800 customers as the training data, and last 200 customers as the test data. The sensitive attributes are gender, and marital-status.

- **Bank Dataset**:[2] Bank dataset contains the information of individuals contacted by a Portuguese bank institution. The assigned classification task is to predict whether the client will subscribe a term deposit. For the classification task we consider all 17 attributes (except martial status as the sensitive attribute). Removing the sensitive attribute, and train a logistic regression model on the dataset, yields to a solution that is biased under the demographic parity notion ($p\% = 70.75\%$). To evaluate the performance of the classifier, we split data into the training (32000 data points), and test set (13211 data points). For the clustering task, we sampled 3 continuous features: Age, balance, and duration. The sensitive attribute is the marital status of the individuals.

- **Adult Dataset**:[3] Adult dataset contains the census information of individuals including education, gender, and capital gain. The assigned classification task is to predict whether a person earns over 50K annually. The train and test sets are two separated files consisting of 32000 and 16000 samples respectively. We consider gender and race as the sensitive attributes (For the experiments involving one sensitive attribute, we have chosen gender). Learning a logistic regression model on the training dataset (without imposing fairness) shows that only 3 features out of 14 have larger weights than the gender attribute. Note that removing the sensitive attribute (gender), and retraining the model does not eliminate the bias of the classifier. the optimal logistic regression classifier in this case is still highly biased ($p\% = 31.49\%$ ). For the clustering task, we have chosen 5 continuous features (Capital-gain, age, fnlwgt, capital-loss, hours-per-week), and 10000 samples to cluster. The sensitive attribute of each individual is gender.

## E    SUPPLEMENTARY FIGURES

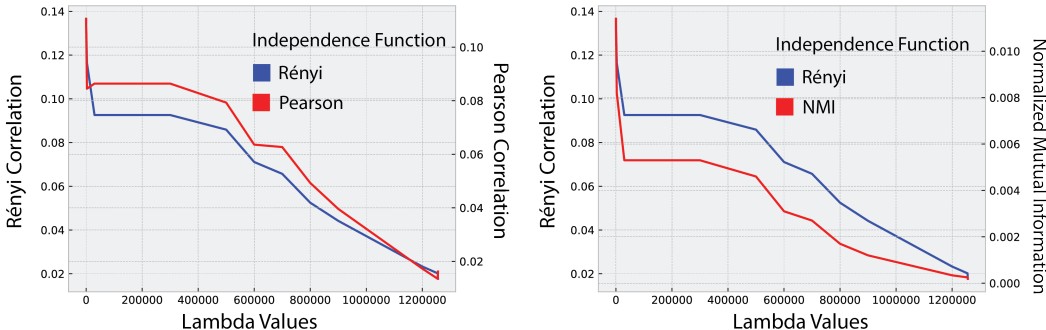

Figure 6: The relationship between Rényi correlation, Pearson correlation, and normalized mutual information. Direct optimization of normalized mutual information is intractable due to its non-convexity. However, as we can observe on the right-hand-side, by minimizing Rényi correlation to 0, the normalized mutual information is converging to 0 accordingly.

---

[1] https://archive.ics.uci.edu/ml/datasets/statlog+(german+credit+data)

[2] https://archive.ics.uci.edu/ml/datasets/Bank%20Marketing.

[3] https://archive.ics.uci.edu/ml/datasets/adult.

# F    FAIR NEURAL NETWORK

In this section, we train a 2-layers neural network on the adult dataset regularized by the Rényi correlation. In this experiment, the sensitive attribute is gender. We set the number of nodes in the hidden layer, the batch-size, and the number of epochs to 12, 128, and 50, respectively. The following table depicts the performance of the trained model.

| $p\%$ | Test Accuracy | Time (Seconds) |
|---|---|---|
| 31.49% | 85.33 | 731 |
| 80.42% | 83.34 | 915 |

Table 1: Performance and training time of a neural network trained on the Adult dataset. The first and second rows correspond to the networks not regularized, and regularized by Rényi correlation respectively. As can be seen in the table, while adding Rényi regularizer makes the classifier more fair, it does so by bringing a minimum amount of additional computational overhead.

