# OpenReview forum: "Rényi Fair Inference"
_ICLR.cc/2020/Conference — Accept (Poster)_

### Official Review · AnonReviewer3 · 2019-10-24
**Official Blind Review #3**

**Rating:** 8

**Review:**

This paper proposes a novel approach to fair inference/learning by using the Renyi correlation in place of the standard correlation measures (e.g. Pearson correlation) as a measure of "unfairness" expressed as a form of dependence between an outcome and a sensitive variable (which includes a number of standard group fairness definitions including demographic parity and equality of odds). This is motivated by the limitation of standard notions of correlation that they only capture linear correlations.
The proposed approach is a formulation using the Renyi correlation as a regularization term, which leads to a min-max formulation due to the definition of Renyi correlation as the maximum correlation between any functionals of the variables in question.
The paper then describes in detail how the computation can be done by making use of the fact that Renyi correlation can be computed using singular values of the relevant matrix, and provides computation time analyses for a couple of different assumptions.
The authors also apply the analogous idea to (k means) clustering to derive what they call the Renyi fair clustering.
In the experimental evaluation section, they evaluate the performance of the proposed methods with respect to the accuracy fairness trade-off, using 3 publicly available real world data sets and with the 2 standard definition of group fairness. The experimental results show quite convincingly that the trade of is improved by Renyi correlation as compared to the use of standard correlations.  They also evaluate the Renyi fair clustering method using a fairness metric they propose for evaluating the uniformity of clustering results.
Overall the paper presents a crisp new idea, develops sound theoretical analysis and algorithms, and validate their superiority with satisfactory experiments.

**Experience Assessment:**

I have read many papers in this area.

**Review Assessment: Checking Correctness Of Derivations And Theory:**

I assessed the sensibility of the derivations and theory.

**Review Assessment: Checking Correctness Of Experiments:**

I assessed the sensibility of the experiments.

**Review Assessment: Thoroughness In Paper Reading:**

I read the paper at least twice and used my best judgement in assessing the paper.

---

> ### Author Response · Authors · 2019-11-14
> **Response to Reviewer #3**
>
> We thank the reviewer for the positive assessment of our paper. We have updated several sections of the paper as follows:
> - A new section conclusion is added to summarize the contribution of the paper and show future direction.
> - Supplementary section D is updated to give a more comprehensive description of the datasets used in experiments.
> - A new supplementary section F is added in which we train a 2-layer neural network on the adult dataset.

---

### Official Review · AnonReviewer2 · 2019-10-25
**Official Blind Review #2**

**Rating:** 6

**Review:**

This paper proposes a method to use renyi correlation to improve fairness of ML models by reducing the dependence of outputs wrt sensitive inputs such as gender. Previous models either use a linear dependence measure or use more complex optimization objectives; this paper improves upon them. They first build a min-max objective where the goodness-of-fit and fairness are jointly optimized. This non-convex objective is difficult to optimize and the authors propose a reformulation of the renyi correlation for discrete random variable case where it reduces to finding second largest eigenvalue. Based on this, they reformulate the objective which can be optimized more efficiently. They show the performance of their model for supervised and unsupervised learning problems on 4 different dataset by comparing to standard correlations such as Pearson.

Overall the paper is clearly written and I liked the idea of using renyi correlation which also has a nice theoretical formulation allowing to be optimized more efficiently. But, the experimental results are a bit weak since the only model they experiment with is logistic regression. Given that their main motivation is to capture non-linear dependencies, I think some results with neural networks is necessary. Since their main focus is on discrete case, the authors can show if training a word embedding model with renyi regularization helps improve fairness of word embeddings.

I have several question regarding the paper:

The datasets that authors use have a predefined feature space. Can you show if the sensitive feature is really important for high accuracy? Can we get the same performance without the sensitive features?
Since the model is trained with gradient descent, how would a more simple baseline where the gradient of the sensitive feature is penalized work?
How would this algorithm generalize to larger problems such as language models since Q_{theta} is regenerated at every iteration?

**Experience Assessment:**

I have read many papers in this area.

**Review Assessment: Checking Correctness Of Derivations And Theory:**

I assessed the sensibility of the derivations and theory.

**Review Assessment: Checking Correctness Of Experiments:**

I carefully checked the experiments.

**Review Assessment: Thoroughness In Paper Reading:**

I read the paper thoroughly.

---

> ### Author Response · Authors · 2019-11-14
> **Response to Reviewer #2**
>
> First, we thank the reviewer for his/her valuable assessment of our paper. Below, we address the main concerns of the reviewer and mention the revisions we have made on the paper.
>
> [Fair word embeddings]
> We appreciate the reviewer for bringing this point up. The notions of fairness and the proposed optimization problems covered in this paper are based on the availability of explicit labels for the sensitive attribute. Hence, it is not straightforward to directly apply this framework to promoting fairness in the training of word embeddings unless there is an explicit attribute such as gender associated with every word that is to be protected. This remains as a natural direction for future exploration, and we have added a line in the conclusion section about it.
>
> [More complex models]
> We have added experiments on more complex models and problems to capture the effectiveness of the proposed Renyi fair inference. We trained a 2-layer neural network model on the adult dataset (32000 training points), where the batch-size = 128, and the number of epochs = 50. As discussed in the supplementary section F, imposing fairness, in this case, increases the training time from 12 minutes to 15 minutes. This shows the effectiveness of the framework when used alongside more complex models in practice.
>
> [Impact of sensitive features on performance and fairness] We have updated supplementary section D to address this comment. For instance, for the adult dataset, the logistic regression model (without imposing fairness) is highly biased against women (p% = 31%). If we include gender during the training phase, at the end of the day, the absolute value of the corresponding weight will be bigger than the corresponding weights of 10 features out of 14.  Note that even if we exclude gender (as the sensitive attribute) during training, this will not make much of a difference (see section D). On the other hand, we see Renyi fair inference consistently achieves high fairness regardless of whether or not the sensitive attribute is included.
> To demonstrate this, we compare four different cases: Fair training (or not) + sensitive attribute included in training (or not), reported here.
>
> Test error and p% when gender included, no fairness regularizer: 85.29%, 32.04%
> Test error and p% when gender not included, no fairness regularizer: 85.15%, 32.02%
> Test error and p% when gender included, adding Renyi regularizer: 82.96%, 99.13%
> Test error and p% when gender not included, adding Renyi regularizer: 82.96%, 99.13%
>
>
> [Adding the gradient of the sensitive feature] We penalize the objective function if the Renyi correlation between the prediction vector and the sensitive attribute(s) vector is large. It is not clear to us how adding the norm of the gradient of the objective function with respect to the sensitive attribute can help making the model fair and provide guarantees as we do in this paper. Please let us know if we are misunderstanding your comment.

---

### Official Review · AnonReviewer1 · 2019-10-31
**Official Blind Review #1**

**Rating:** 6

**Review:**

This work shows an interesting approach to introduce fairness to machine learning. The introduction and presenation of the approach are well written. The introduction provides a valuable overview and characterization of existing work and motivates the approach proposed in this work. Sections 2 through 5, together with the supplementary material provide a comprehensible description and derivation of the approaches to classification and clustering, together with the corresponding algorithms.
However, experimental section leaves open a number of questions. First of all, for readers not familiar with the task used here, a more detailed description would be desirable, including moving Supplementary Section D into the main text. In Suppl. Sec. D also a succinct definition of the classification task should be provided. What exactly are the classiffication tasks you solve for the Bank and Adult dataset, i.e. what is the set (and number) of classes in both cases?
Figs. 1 (a) and (b): you should provide a legend defining the red and blue curves.
Fig 1 (c): without knowing the other approaches cited here, some comment on the differences in the classifier models used would be helpful to evaluate the differences in test error. Effectively, the reader would like to get an indication, what the differences in test error are induced by: difference in the regulatory fairness terms, or also differences in the classifier models.
Overall, the three datasets used to test the approach seem to be fairly limited, as are the corresponding classifier models used. It would be interesting to get an indication of how the approach would perform on/scale to much larger tasks with (much) deeper classification models?
Completely missing is a final overall conclusion section.

**Experience Assessment:**

I do not know much about this area.

**Review Assessment: Checking Correctness Of Derivations And Theory:**

I assessed the sensibility of the derivations and theory.

**Review Assessment: Checking Correctness Of Experiments:**

I carefully checked the experiments.

**Review Assessment: Thoroughness In Paper Reading:**

I read the paper at least twice and used my best judgement in assessing the paper.

---

> ### Author Response · Authors · 2019-11-14
> **Response to Reviewer #1**
>
> We thank the reviewer for constructive feedback. Below, we respond to the questions and explain the changes that have been made to the paper based on your comments.
>
> [More detailed description in the numerical experiments] We have expanded the legends of the figures and the description of the experiments. While we expanded Appendix D, we decided not to move it to the main body due to space limitations.
>
>
> [Fig. legends] We have expanded the legends for all figures so that they are now all self-explanatory.
>
> [Baselines in Fig. 1 (c)] We added a paragraph explaining the differences between the classifiers used as benchmarks in Fig 1 (c) (second paragraph of Section 6) and the reasons why Renyi fair classification outperforms them.
>
> [More complex models]
> We have added experiments on more complex models and problems to capture the effectiveness of the proposed Renyi fair inference. We trained a 2-layer neural network model on the adult dataset (32000 training points), where the batch-size = 128, and the number of epochs = 50. As it is discussed in supplementary section F, imposing fairness in this case, increases the training time from 12 minutes to 15 minutes. This showcases the effectiveness of the framework when used alongside more complex models in practice.
>
>
> [Scalability of Renyi Classifier] To understand the computational overhead introduced to the algorithm for imposing fairness, let us compare the Renyi fair classification approach to Non-fair-GD method. Non-fair-GD method is the gradient descent method applied to the empirical risk minimization problem with no regularization or constraint for imposing fairness. Let us start by comparing Algorithm 2 with its non-fair-GD counterpart. First of all, notice that Algorithm 2 requires an extra step (Step 3 in Algorithm 2) in each iteration compared to non-fair-GD. The computational complexity of step 3 is O(n) and is the same as computing the gradient in step 4. Thus the per-iteration computational complexity of Algorithm 1 is order-wise the same as its non-fair-GD counterpart. Moreover, as discussed in the revised paper after Theorem 4.2, the number of iterations required for convergence of Algorithm 2 matches the iteration complexity lower-bound. Therefore, the number of iterations required for convergence of Algorithm 2 is order-wise the same as its non-fair-GD counterpart in non-convex smooth cases. This observation shows that overall, we only lose in constants (if at all losing in terms of computational complexity).
>
> Now, let us compare Algorithm 1 with its non-fair-GD counterpart. In this case, step 3 in Algorithm 1 requires finding the second singular vector of the matrix Q. The dimension of this matrix depends on the number of classes in the classification and the number of categories in the sensitive attribute. Therefore, the dimension of matrix Q does not scale with the number of data points or the dimension of the model; and therefore in many cases remains small. For example, as a case study, for a problem that has 10 target classes, and 4 possible values for the sensitive attributes (white-man, white-woman, black-man, black woman as an example) finding the second largest eigenvector of a 10*4 matrix is not computationally expensive. Notice that for such a matrix the first singular vector is known (see v_1 definition after equation (6)). Thus one can subtract the component corresponding to the largest singular vector and use fast methods such as power method to find the second singular vector. This shows that practically the per-iteration computational complexity of Algorithm 1 is no more than its non-fair-GD counterpart. However, the total number of iterations (see Theorem 4.1) is larger than the total number of required iterations for convergence in the non-fair-GD counterpart. This can be viewed as the computational cost of imposing fairness in this case.
>
>
> [Conclusion section] We have now written a conclusion section summarizing our contributions and findings.

---

### Decision · Program_Chairs · 2019-12-19

**Decision:**

Accept (Poster)

**Comment:**

The paper addresses the problem of fair representation learning. The authors propose to use Rényi correlation as a measure of (in)dependence between the predictor and the sensitive attribute and developed a general training framework to impose fairness with theoretical properties. The empirical evaluations have been performed using standard benchmarks for fairness methods and the SOTA baselines -- all this supports the main claims of this work's contributions.
All the reviewers and AC agree that this work has made a valuable contribution and recommend acceptance. Congratulations to the authors!